# Noise Modeling and Simulation of Giant Magnetic Impedance (GMI) Magnetic Sensor

**DOI:** 10.3390/s20040960

**Published:** 2020-02-11

**Authors:** Fang Jin, Xin Tu, JinChao Wang, Biao Yang, KaiFeng Dong, WenQin Mo, YaJuan Hui, JunWen Peng, JieFeng Jiang, Lei Xu, JunLei Song

**Affiliations:** 1School of Automation, China University of Geosciences, Wuhan 430074, China; 2Hubei key Laboratory of Advanced Control and Intelligent Automation for Complex Systems, Wuhan 430074, China

**Keywords:** GMI sensor, noise modeling, 1/f noise, white noise, equivalent input magnetic noise

## Abstract

The detection resolution of a giant magneto-impedance (GMI) sensor is mainly limited by its equivalent input magnetic noise. The noise characteristics of a GMI sensor are evaluated by noise modeling and simulation, which can further optimize the circuit design. This paper first analyzes the noise source of the GMI sensor. It discusses the noise model of the circuit, the output sensitivity model and the modeling process of equivalent input magnetic noise. The noise characteristics of three modules that have the greatest impact on the output noise are then simulated. Finally, the simulation results are verified by experiments. By comparing the simulated noise spectrum curve and the experimental noise spectrum curve, it is demonstrated that the preamplifier and the multiplier contribute the most to the output white noise, and the low-pass filter plays a major role in the output 1/f noise. These modules should be given priority in the optimization of the noise of the conditioning circuit. The above results provide technical support for the practical application of low-noise GMI magnetometers.

## 1. Introduction

Since the discovery of the giant magneto-impedance (GMI) effect in the CoFeSiB soft magnetic material by the Mohri team in Japan in 1992, the GMI effect has been found in many soft magnetic materials [1,2]. The magnetic field sensitivity can reach 2–1000%/Oe, and it has been paid more and more attention by researchers all over the world [3,4,5,6].

Noise characteristics have always been a key indicator that determines the resolution of magnetic sensors [7,8]. According to previous research results, the noise of a GMI sensor mainly comes from three sources: GMI component intrinsic noise, conditioning circuit noise and external interference noise [9]. The noise testing experiment of the sensor is carried out in the environment of a shielding cylinder, the purpose of which is to shield the external interference noise. The intrinsic noise of GMI components is the effect of multiple noise sources. In the middle and high frequencies (more than 1 kHz), Barkhausen noise is the mainstay, and it is caused by the complex magnetic domain structure forming the ferromagnetic material and the displacement and rotation of the domain wall during magnetization [10,11]. In the low frequencies (less than 1 kHz), the thermomagnetic noise caused by the thermal fluctuation of magnetization contributes significantly to the intrinsic noise of the GMI component through impedance fluctuations [12,13]. The intrinsic noise of the GMI component is 2~3 magnitudes [14] lower than that of the conditioning circuit. Above all, the conditioning circuit is the main factor of noise sources [15,16,17], and it is mainly studied in this paper.

Conditioning circuit noise is inherent noise inside an electronic system, which is caused by the random motion of the charge carriers. It includes the thermal noise of the resistor [18,19], the shot noise of the PN junction and the 1/f noise [20,21]. For the GMI sensor, the conditioning circuit structure is divided into two parts: the excitation circuit and the detection circuit. The excitation circuit mainly includes an incentive source and voltage-to-current converter [22]. The detection circuit includes a preamplifier, a peak detection circuit and an instrumentation amplifier [23,24]. Each part contains multiple noise sources, and the noise effect contributes differently to the total output noise of the sensor. According to the Fries theorem, in order to achieve the best noise characteristics, the signal-to-noise ratio and the equivalent input noise voltage are generally used to weigh and design the various parts of the system [25,26]. Therefore, by modeling the noise of each module in MATLAB, the dominant noise source is found, allowing further optimization of the dominant noise source and significantly improving the noise characteristics of the conditioning circuit.

The work of this paper is divided into three steps: firstly, the modeling idea of the equivalent input magnetic noise model of a GMI sensor is discussed. The output voltage noise model, sensitivity model and equivalent input magnetic noise model are established. Then the noise contribution of each module is computed, and the optimization scheme of the dominant noise source is discussed. Finally, the effectiveness of the noise optimization method is verified by the noise test experiment, and the characteristics of the low-noise GMI sensor conditioning circuit are summarized. This work provides theoretical support for the design of low-noise GMI sensors.

## 2. GMI Sensor Noise Modeling

The GMI sensor is composed of seven parts: A. Excitation source; B. Voltage-to-current converter; C. GMI components; D. Preamplifier; E. Multiplier; F. Filter and G. Instrumentation amplifier. The output voltage of the instrumentation amplifier is used as the output signal of the GMI sensor. The schematic is shown in Figure 1.

For the above GMI sensor circuit, the establishment process of the equivalent input magnetic noise model is carried out in three steps. First, according to the superposition theorem, the output voltage noise model of the sensor is established. Using the noise source model of the electronic components, the internal noise model of each module is established as well. Further, considering the impact of power gain, the contribution of each module noise to the total output voltage noise is obtained. Second, the sensitivity model is established based on the inherent sensitivity of the GMI component, the excitation voltage amplitude and the gain of the conditioning circuit. Third, an equivalent input magnetic noise model is established from the above two models.

### 2.1. Noise Model of the Output Voltage

According to the schematic circuit diagram in Figure 1, the noise power and power gain of each module of the GMI sensor circuit are shown in Table 1.

In Table 1, $E_{nX}^{2}$ represents the sum of the noise power generated inside the module *X*. The output of this module is also the input noise of the subsequent modules. $G_{X}^{2}$ is the power gain function corresponding to the module, representing the gain of the input noise power, and X represents each module. The next step is to solve the noise of each module.

#### 2.1.1. Excitation Source

The Direct Digital Synthesis (DDS) excitation source used in this paper is AD9959. The source of noise is mainly caused by phase noise and is characterized by noise spectral density (NSD). The output noise power $E_{ng}^{2}$ of the excitation source is calculated as:(1)$$E_{ng}^{2}\;=\;{\left(\frac{V_{g}/\sqrt{2}}{10^{148/20}}\right)}^{2}\;=\;{\left(\frac{V_{g}}{10^{151/20}}\right)}^{2}$$
where $V_{g}$ represents the peak-to-peak value of the output signal of the excitation source. The phase noise is attenuated for the carrier frequency, and its value is −148dbc/Hz.

#### 2.1.2. Voltage-to-Current Converter

The function of the voltage-to-current converter is to convert the high-frequency alternating voltage signal into an alternating current signal. This alternating current signal is used as an excitation for the GMI component. According to Norton’s theorem, the converter is equivalent to a current source, and its power gain function $G_{vi}^{2}$ is expressed as:(2)$$G_{vi}^{2}\;=\;{\left(\frac{R_{2}}{R_{1}}\cdot \frac{1}{R_{5}}\right)}^{2}$$

The internal noise sources of the subsequent modules are divided into the thermal noise power of each resistor and the equivalent noise power of the Operational amplifier (op-amp). The internal noise source of the Voltage-to-current converter is divided into two parts.

##### The Thermal Noise Power of Resistor

The thermal noise spectral density of each resistor is expressed as $E_{ti}^{2}$, I = 1~14, representing the thermal noise spectral density of resistors $R_{1}$~$R_{14}$ respectively. The contribution of the Voltage-to-current converter at the output is $E_{tvi}^{2}$.
(3)$$E_{tvi}^{2}\;=\;E_{t1}^{2}{\left(\frac{R_{2}}{R_{1}}\cdot \frac{1}{R_{5}}\right)}^{2}\;+\;E_{t2}^{2}{\left(\frac{1}{R_{5}}\right)}^{2}\;+\;E_{t3}^{2}{\left(\frac{R_{2}}{R_{1}}\cdot \frac{1}{R_{5}}\right)}^{2}\;+\;E_{t4}^{2}{\left(\frac{1}{R_{5}}\right)}^{2}\;+\;E_{t5}^{2}{\left(\frac{1}{R_{5}}\right)}^{2}$$
when $R_{1}\;=\;R_{3}$, $R_{2}\;=\;R_{4}$, it is simplified to Equation (4).
(4)$$E_{tvi}^{2}\;=\;2E_{t1}^{2}{\left(\frac{R_{2}}{R_{1}}\cdot \frac{1}{R_{5}}\right)}^{2}\;+\;2E_{t2}^{2}{\left(\frac{1}{R_{5}}\right)}^{2}\;+\;E_{t5}^{2}{\left(\frac{1}{R_{5}}\right)}^{2}$$

##### Equivalent Noise Power of the Op-Amp

The equivalent noise of the op-amp is calculated by the $\left(e_{n},\;i_{n}\right)$ model, where $e_{nX}$ represents the equivalent input voltage noise spectral density, $i_{nX}$ represents the equivalent input current noise spectral density, and *X* represents each module.
(5)$$E_{vvi}^{2}\;=\;{\left(\frac{R_{1}\;+\;R_{2}}{R_{1}}\cdot \frac{1}{R_{5}}\right)}^{2}\cdot e_{nvi}^{2}\;+\;{\left(\frac{1}{R_{5}}\right)}^{2}\cdot e_{nvi}^{2}$$
(6)$$E_{ivi}^{2}\;=\;2\cdot {\left(R_{2}\cdot \frac{1}{R_{5}}\right)}^{2}\cdot i_{nvi}^{2}\;+\;i_{nvi}^{2}$$

The contribution of the voltage source noise of the two op-amps at the output is expressed as $E_{vvi}^{2}$ in Equation (5). The contribution of the current source noise at the output is expressed as $E_{ivi}^{2}$ in Equation (6). Thus, the total output noise power of the Voltage-to-current converter is expressed as follows.
(7)$$E_{nvi}^{2}\;=\;|Z\left(H_{ex}\right){|}^{2}\cdot \left(E_{tvi}^{2}\;+\;E_{vvi}^{2}\;+\;E_{ivi}^{2}\right)$$
where $Z\left(H_{ex}\right)$ represents the impedance length value of the GMI component.

#### 2.1.3. Preamplifier

The preamplifier acts as a buffer, which is used to collect the voltage across the GMI component. To reduce the effect on the voltage of the GMI component during access, the input impedance should be large enough, so the preamplifier is built with a non-inverting amplifier. Its power gain function is $G_{pre}^{2}$.
(8)$$G_{pre}^{2}\;=\;{\left(1\;+\;\frac{R_{9}}{R_{8}}\right)}^{2}$$

The internal noise source of the preamplifier is divided into the thermal noise of each resistor and the equivalent noise of the input of the op-amp.

##### The Thermal Noise Power of Resistor

This part includes resistor $R_{8},\;R_{9}$, and the equivalent resistance of the GMI component. The resistance thermal noise power is $E_{tpre}^{2}$.
(9)$$E_{tpre}^{2}\;=\;E_{tGMI}^{2}{\left(1\;+\;\frac{R_{9}}{R_{8}}\right)}^{2}\;+\;E_{t8}^{2}{\left(\frac{R_{9}}{R_{8}}\right)}^{2}\;+\;E_{t9}^{2}\cdot 1$$
where $E_{tGMI}$ represents the thermal noise of the equivalent resistance of the GMI component, $E_{t8}$ represents the thermal noise of the input resistor $R_{8}$ at the inverting terminal, and $E_{t9}$ represents the thermal noise of the feedback resistor $R_{9}$.

##### Equivalent Noise Power of the Op-Amp

The equivalent noise power of the op-amp is obtained according to the $\left(e_{n},\;i_{n}\right)$ model.
(10)$$E_{vpre}^{2}\;=\;{\left(1\;+\;\frac{R_{9}}{R_{8}}\right)}^{2}\cdot e_{npre}^{2}$$
(11)$$E_{ipre}^{2}\;=\;R_{9}^{2}\cdot i_{npre}^{2}\;+\;{\left(1\;+\;\frac{R_{9}}{R_{8}}\right)}^{2}R_{GMI}^{2}\cdot i_{npre}^{2}$$
where $R_{GMI}$ represents the equivalent resistance of the GMI component.

The total output noise power of the preamplifier $E_{npre}^{2}$ is expressed as:(12)$$E_{npre}^{2}\;=\;E_{tpre}^{2}\;+\;E_{vpre}^{2}\;+\;E_{ipre}^{2}$$

#### 2.1.4. Multiplier

The lock-in amplifier is made up by multiplier and low-pass filter, which achieve the amplitude detection of input signal. In this process, the reference signal is input by the DDS source, the noise of source is mainly caused by phase noise, which is smaller than the signal under test. After the signal under test passes through the multiplier, the output noise power becomes $\sqrt{2}$ times of the input noise power, and the power gain of the multiplier is expressed as:(13)$$G_{nmul}^{2}\;=\;{\left(\sqrt{2}\cdot G_{mul}\right)}^{2}$$
where $\sqrt{2}$ is the noise power gain factor. $G_{mul}$ is the voltage gain of the multiplier, which is expressed as:(14)$$G_{mul}\;={\;V}_{r}\cdot K$$
where $V_{r}$ represents the reference signal amplitude, K represents the gain coefficient of the multiplier, whose value is 0.5 cosθ, and its unit is 1/V. θ represents the phase difference between the signal under test and the reference signal.

Apart from the gain on the input noise, the multiplier itself generates noise, which is expressed as an internal equivalent output noise source $E_{mul}$ So the total output noise power of the multiplier $E_{nmul}^{2}$ is expressed as the following:(15)$$E_{nmul}^{2}\;=\;E_{mul}^{2}$$

#### 2.1.5. Filter

The filter is a 2nd order Butterworth low-pass filter. By filtering the signal output of the multiplier, the DC component of the signal is obtained. Power gain of the filter is a function of frequency, which is expressed as:(16)$$G_{filter}^{2}\;=\;{\left(\frac{1}{\sqrt{1\;+\;\left(f/f_{c}\right)}}\right)}^{2}$$
where $f_{c}$ is the cutoff frequency of the low-pass filter, it is determined by the values of resistance and capacitance of the filter.

The internal noise source of the filter is divided into the thermal noise power of each resistor and the equivalent noise power of the op-amp.

##### The thermal noise power of resistor

(17)$$E_{tfilter}^{2}\;=\;E_{t12}^{2}+E_{t13}^{2}$$

##### Equivalent noise power of the op-amp

(18)$$E_{vfilter}^{2}\;=\;e_{nfilter}^{2}\cdot 1$$

(19)$$E_{ifilter}^{2}\;=\;{\left(R_{12}\;+\;R_{13}\right)}^{2}\cdot i_{nfilter}^{2}$$

Therefore, the total output noise power of the low-pass filter $E_{nfilter}^{2}$ is expressed as:(20)$$E_{nfilter}^{2}\;=\;E_{tfilter}^{2}\;+\;E_{vfilter}^{2}\;+\;E_{ifilter}^{2}$$

#### 2.1.6. Instrumentation Amplifier

A zero-amplifier circuit is built based on the instrumentation amplifier, which realizes zero adjustment and post-amplification of the output of the filter. Its power gain $G_{INA}^{2}$ is set by the gain resistor $R_{gain}$.
(21)$$G_{INA}^{2}\;=\;{\left(1\;+\;\frac{50000}{R_{gain}}\right)}^{2}$$

The internal noise source of the instrumentation amplifier is divided into the thermal noise of each resistor and the equivalent noise of the input of the op-amp.

##### The thermal noise power of resistor

Here we mainly consider the zero-potentiometer thermal noise, which is expressed as:(22)$$E_{tINA}^{2}\;=\;G_{INA}^{2}\cdot E_{t14}^{2}$$

##### Equivalent noise power of the op-amp

It is worth noting that, unlike the commonly used single op-amp structure, the instrument op-amp INA128 consists of three op-amps, which are divided into 2 levels. The first two op-amps serve as input buffers of the non-inverting and inverting inputs. The third op-amp builds a differential amplifier circuit, which realizes the amplifier circuit with high input impedance and high common mode rejection ratio. The noise model is described by the $\left(e_{n},\;i_{n}\right)$ model. The equivalent voltage and equivalent current noise power are expressed as:(23)$$E_{vINA}^{2}=G_{INA}^{2}\cdot e_{nINA}^{2}$$
(24)$$E_{iINA}^{2}=G_{INA}^{2}\cdot R_{14}^{2}\cdot i_{nINA}^{2}$$

Therefore, the total output noise power of the instrumentation amplifier $E_{nINA}^{2}$ is expressed as:(25)$$E_{nINA}^{2}\;=\;G_{INA}^{2}\left(E_{tINA}^{2}\;+\;E_{vINA}^{2}\;+\;E_{iINA}^{2}\right)$$

In summary, in the GMI sensor signal conditioning circuit, the noise expression of each module and its power gain expression have been given. The system block diagram of the noise power and gain transfer function is shown in Figure 2.

It is worth noting that there are differences between the gain transfer functions in the system. For linear systems, such as voltage-to-current converters and preamplifiers, the expression is the quadratic of the voltage gain. For nonlinear systems, such as multipliers, the power gain is $\sqrt{2}$ times the voltage gain, which is related to the amplitude–frequency characteristic of the multiplier.

The expression of the total output noise power $E_{ntotal}^{2}$ of the GMI sensor is calculated as:(26)$$\begin{array}{c} E_{ntotal}^{2}\;=\;E_{ng}^{2}\left(Z{\left(H_{ex}\right)}^{2}\cdot G_{vi}^{2}\cdot G_{pre}^{2}\cdot 2G_{mul}^{2}\cdot G_{filter}^{2}\cdot G_{INA}^{2}\right)+\;\\ \;E_{nvi}^{2}\left(G_{pre}^{2}\cdot 2G_{mul}^{2}\cdot G_{filter}^{2}\cdot G_{INA}^{2}\right)+\\ E_{npre}^{2}\left(2G_{mul}^{2}\cdot G_{filter}^{2}\cdot G_{INA}^{2}\right)+\\ \;E_{nmul}^{2}\left(G_{filter}^{2}\cdot G_{INA}^{2}\right)+\\ \;E_{nfilter}^{2}\cdot G_{INA}^{2}\;+\;E_{nINA}^{2} \end{array}$$

The output voltage noise model of the conditioning circuit has been introduced. We will discuss the sensitivity model of the sensor as follows.

### 2.2. Sensitivity Model

For GMI sensors, the output voltage sensitivity $S_{v}$ is defined as
(27)$$S_{v\;}=\;\frac{dV_{out}}{dH_{ex}}$$
where $V_{out}$ represents the output voltage of the GMI sensor, and $H_{ex}$ represents the external magnetic field, thereby the unit of $S_{v}$ is V/T.

Based on the conditioning circuit of the GMI sensor, the expression of the output voltage sensitivity is expressed as
(28)$$S_{v}\;=\;S_{\Omega }\cdot I_{g}\cdot \left(G_{pre}G_{mul}G_{filter}G_{INA}\right)$$

Here $S_{\Omega }$ represents the inherent sensitivity of the GMI component. $I_{g}$ is the magnitude of the excitation current flowing through the GMI component. $G_{X}$ represents the gain of each conditioning circuit blocks. *X* represents each module.

The magnitude of the $I_{g}$ is related to the amplitude of the excitation source voltage and the gain of the voltage-to-current converter, which is expressed as:(29)$$I_{g}\;=\;V_{g}\cdot G_{vi}$$

Combining Equations (28) and (29), the GMI sensor output voltage sensitivity expression is
(30)$$S_{v}\;=\;S_{\Omega }\cdot V_{g}\cdot \left(G_{vi}G_{pre}G_{mul}G_{filter}G_{INA}\right)$$

### 2.3. Model of Equivalent Input Magnetic Noise

By analyzing the above two models, the equivalent input magnetic noise level $B_{ntotal}$ of the GMI sensor is inferred as:(31)$$B_{ntotal}\;=\;\frac{E_{ntotal}}{S_{v}}$$
where $E_{ntotal}$ represents the output voltage noise spectral density of the GMI sensor, and its unit is V/√Hz. From this, the unit of the equivalent input magnetic noise spectral density $B_{ntotal}$ is T/√Hz.

Combining Equations (26), (30) and (31), the model of equivalent input magnetic noise power is inferred as:(32)$$\begin{array}{c} B_{ntotal}^{2}\;=\;\frac{1}{S_{\Omega }^{2}\cdot V_{g}^{2}}[{\left(E_{ng}\cdot Z\left(H_{ex}\right)\cdot \sqrt{2}\right)}^{2}\;+\;{\left(\frac{E_{nvi}}{G_{vi}}\cdot \sqrt{2}\right)}^{2}\;+\;{\left(\frac{E_{npre}}{G_{vi}\cdot G_{pre}}\cdot \sqrt{2}\right)}^{2}\;+\;\\ {\left(\frac{E_{nmul}}{G_{vi}\cdot G_{pre}\cdot G_{mul}}\right)}^{2}\;+\;{\left(\frac{E_{nfilter}}{G_{vi}\cdot G_{pre}\cdot G_{mul}\cdot G_{filter}}\right)}^{2}\;+\;\\ {\left(\frac{E_{nINA}}{G_{vi\;}\cdot \;G_{pre\;}\cdot \;G_{mul}\;\cdot \;G_{filter\;\cdot }G_{INA}}\right)}^{2}] \end{array}$$

## 3. Model Simulation Results

According to the output voltage noise model, the output voltage noise spectral density curve of the GMI sensor is obtained. The parameters of the model simulation are set as follows: the amplitude of the excitation voltage $V_{g}$ is 1 V, the gain of the voltage-to-current converter $G_{vi}$ is 0.01 A/V, the excitation current amplitude is 0.01 A, the frequency $f_{g}$ is 5 MHz and the analysis bandwidth is 10 kHz. According to the sensitivity model, the output voltage sensitivity is calculated as 1 × 10^4^ V/T. Finally, the output voltage noise spectral density curve of each module is obtained, as shown in Figure 3.

As shown in Figure 3, the analysis bandwidth is 1 Hz to 10 kHz. For the four modules before modulation and demodulation (A. Excitation source, B. Voltage-to-current converter, D. Preamplifier and E. Multiplier), the main source of noise is white noise. For the two modules after demodulation (F. Filter and G. Instrumentation amplifier), the noise is mainly concentrated in the low frequencies, which is a superposition effect of 1/f noise and white noise. Therefore, the total output noise voltage spectral density is the sum of the output voltage spectral densities of all the individual modules. In the range of white noise (100 Hz~10 kHz), the total output noise $e_{ntotal}$ is 1.39 × 10^−5^ V/√Hz, where the C module ($e_{npre}$ = 7.53 × 10^−6^ V/√Hz) and D module ($e_{nmul}$ = 1 × 10^−5^ V/√Hz) contribute the most noise of the total output, while the noise of the B and F modules has less effect on the total output noise. In the range of 1/f noise (1~100 Hz), the total output noise @1 Hz is $e_{ntotal}\;$= 5.52 × 10^−5^ V/√Hz, and the module F @1 Hz is $e_{nfilter}$ = 5.11 × 10^−5^ V/√Hz. Hence the total output noise is mainly determined by the F module.

Therefore, according to the contribution of each module to the total output voltage noise spectral density, it is not difficult to find that the D and E modules are the main source of output noise in the range of white noise, and the F module is the main source of output noise in the range of 1/f noise. In the following work, according to the noise model of the conditioning circuit, the noise of the D, E and F modules is studied and optimized.

### 3.1. Module D—Preamplifier

The noise model of the preamplifier shows that the noise source includes two parts: the resistance thermal noise and the noise of the op-amp. The simulation parameters are shown in Table 2.

In Table 2, according to the datasheet of the preamplifier, three different configurations are set. For configurations 1 and 2, the op-amps are OPA842, while the values of resistors $R_{8}$ and $R_{9}$ are different. For configuration 2 and configuration 3, the resistors $R_{8}$ and $R_{9}$ have the same value, while the op-amp of configuration 3 is OPA846. The noise contribution in different configurations is shown in Table 3.

As shown in Table 3, the calculation result of configuration 1 shows that the resistance thermal noise contributes the most to the preamplifier noise, while the op-amp voltage noise and current noise do not differ much. By changing the value of $R_{8}$ and $R_{9}$, the result is shown in configuration 2. In comparing of the noise of configurations 1 and 2, the output noise level is significantly reduced in configuration 2, which is mainly limited by the equivalent input voltage noise. If an op-amp with a lower voltage noise is used, as shown in configuration 3, the noise level of the preamplifier is further reduced. Therefore, in the preamplifier module, the selection of resistance and op-amp should be paid more attention.

### 3.2. Module E—Multiplier

From the noise model of the multiplier, the noise mainly comes from the equivalent output noise voltage. It determines the inherent noise level of the multiplier, which is closely related to the multiplier output bandwidth. The carrier frequency $f_{g}$ of the GMI component is 5 MHz. If the bandwidth of the chip is low, it tends to cause nonlinear distortion at the input end. On the contrary, the high bandwidth causes the broadband noise to accumulate in the multiplier, which reduces the signal-to-noise ratio of the multiplier module output. Therefore, the chip’s bandwidth and equivalent input noise need to be fully considered in the selection.

### 3.3. Module F—Low-Pass Filter

In the case of satisfying the cutoff frequency of 200 Hz, the parameters of the resistor and capacitor are set as follows: $R_{12}$ and $R_{13}$ are 10 kΩ, $C_{2}$ is 3.9 μF, $C_{3}$ is 620 nF and the op-amp is selected as the high precision op-amp OPA227. From this, the noise spectral density curve of each noise output voltage in the low-pass filter is calculated as shown in Figure 4.

The filter output total noise $E_{n\mathrm{filter}}$ is the sum of superposition of the resistance thermal noise $E_{nt}$, the operational amplifier voltage noise $E_{nv}$ and the current noise $E_{ni}$. It is seen that the current noise of the op-amp contributes the most noise to the total output noise, especially in the range of 1/f noise. When the frequency f is equal to 1 Hz, $E_{nfilter}$ is 2.56 × 10^−7^ V/√Hz and $E_{ni}$ is 2.53 × 10^−7^ V/√Hz. Hence in the design of low-pass filters, the choice of op-amps is critical.

Above all, we analyzed the contribution of each module to the total output noise level, and then optimized the internal noise source of the dominant module to improve the total output noise level, which would provide a theoretical basis for the latter experimental part.

## 4. GMI Sensor Noise Test Experiment

According to the previous model research, to verify the correctness and feasibility of the circuit design and noise optimization scheme, the GMI magnetic sensor noise test system was built.

As shown in Figure 5, the GMI sensor probe and solenoid are placed in a magnetic shielding cylinder, which can effectively shield the external magnetic field from interference. The generated sensing signal is connected to the spectrum through the sensor conditioning circuit. The voltage noise spectrum of the sensor is obtained by analyzing the output data.

The conditioning circuit’s parameters are the same as the simulation. The output voltage of the instrumentation amplifier is connected to the Spectrum analyzer.

The voltage noise spectral density of the GMI sensor is shown in Figure 6. In the frequency range below 10 Hz, the experimental results are larger than the simulation results, which were caused by ignoring the potentiometer noise in the zeroing circuit. The output noise is dominated by 1/f noise, and the noise level is proportional to the reciprocal of the frequency. Taking the point of 1 Hz as the reference, the total output noise @1 Hz (5.65 × 10^−5^ V/√Hz) is slightly larger than the simulation result (5.52 × 10^−5^ V/√Hz). This is caused by the intrinsic noise of the GMI components.

In the frequency range of 10–1000 Hz, the experimental results agree well with the simulation results, and the measured value is slightly larger than the simulated value because the actual layout and routing of the circuit components were not considered in the simulation. The white noise level of the GMI sensor obtained by the actual test is 1.62 × 10^−5^ V/√Hz.

Further considering the sensitivity of the sensor, it is calculated that the equivalent input magnetic noise level of the GMI sensor is 5.66 nT/√Hz.

## 5. Conclusions

According to the relevant theory of weak signal detection technology, the noise model of a GMI sensor was established. It includes the noise model of the conditioning circuit, the sensitivity model and the equivalent input magnetic noise model. The simulation calculation was carried out by MATLAB, and the influence of conditioning circuit parameters on the noise performance of a GMI sensor was studied. The simulation results show that in the range of white noise, the multiplier module and the preamplifier module contribute the most to the output noise. The equivalent output voltage noise of the multiplier, the resistance thermal noise of the preamplifier and the op-amp’s voltage noise are the dominant noise sources. In the range of 1/f noise, the output noise of the low-pass filter is dominant, and the current noise of the op-amp is the dominant noise source in the low-pass filter. Subsequently, the output voltage noise spectrum curve within the analysis bandwidth of 1000 Hz was measured. Comparing the measured results with the simulation results, it is seen that the two results have good agreement, which indicates that the experiment effectively verified the simulation. The results provide a theoretical basis for the practical application of low-noise GMI magnetic sensors.

This paper only carried out a preliminary experimental verification of the noise simulation model, and further performance improvement will be made based on this model later.

## Figures and Tables

**Figure 1 sensors-20-00960-f001:**
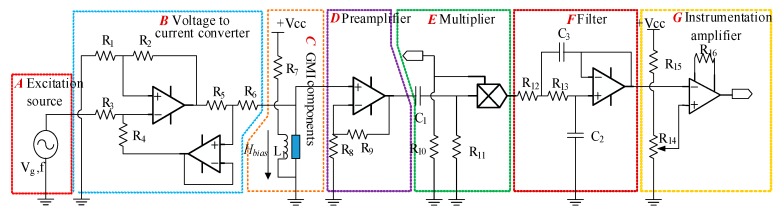
Schematic circuit diagram of the giant magneto-impedance (GMI) sensor.

**Figure 2 sensors-20-00960-f002:**
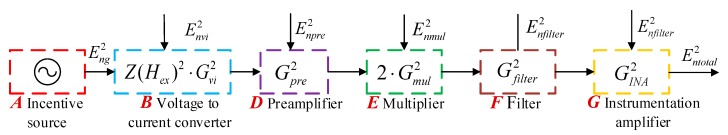
System block diagram of power gain transfer function.

**Figure 3 sensors-20-00960-f003:**
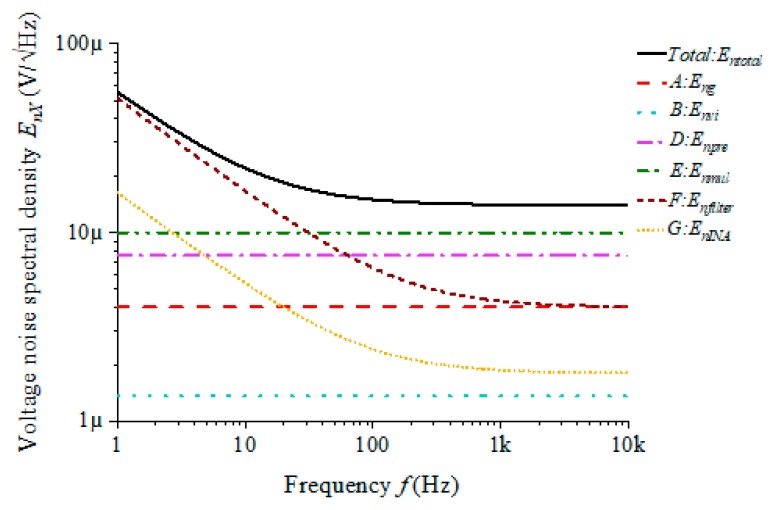
Output voltage noise spectral density of each module: $E_{ntotal}$ is the total output noise, $E_{nX}$ represents the sum of the noise generated inside module X and X represents each module.

**Figure 4 sensors-20-00960-f004:**
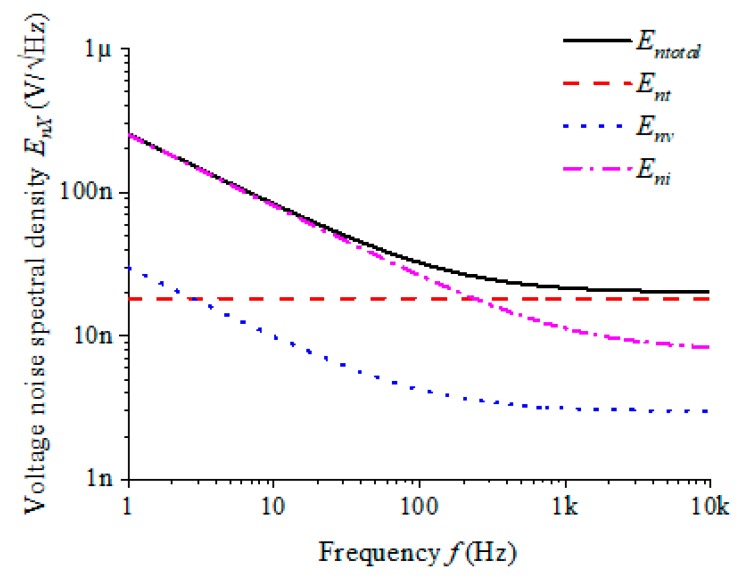
Each Noise contribution of the low-pass filter: $E_{nfilter}$ is the sum of all parts, $E_{nt}$ is the resistance thermal noise, $E_{nv}$ is the operational amplifier voltage noise and $E_{ni}$ is the current noise.

**Figure 5 sensors-20-00960-f005:**
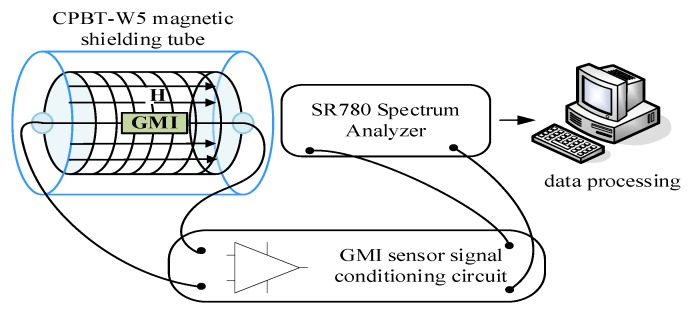
GMI magnetic sensor noise test system diagram.

**Figure 6 sensors-20-00960-f006:**
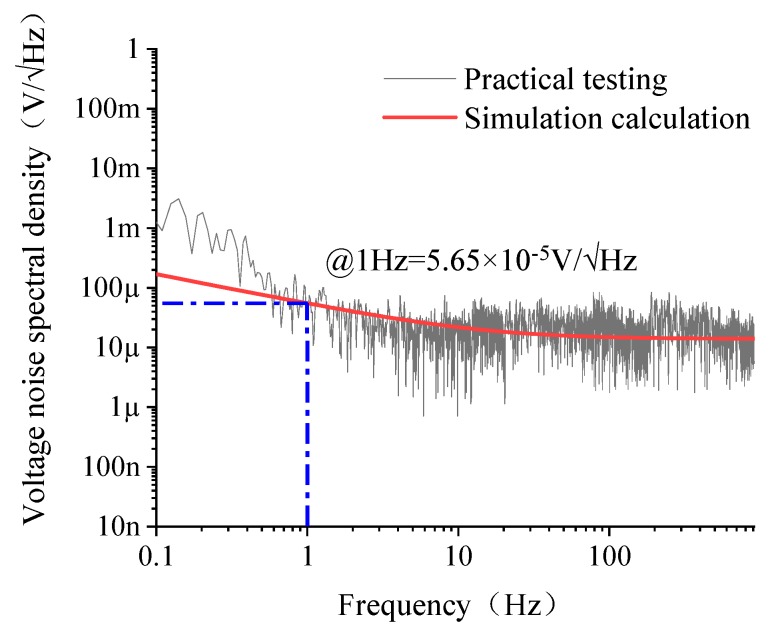
Voltage noise spectral density curve of the GMI sensor.

**Table 1 sensors-20-00960-t001:** Internal noise and power gain description for each module.

Name	Noise Power	Power Gain
A. Excitation source	$E_{ng}^{2}$	*/*
B. Voltage-to-current converter	$E_{nvi}^{2}$	$|Z\left(H_{ex}\right){|}^{2}\cdot G_{vi}^{2}$
C. GMI components	$E_{nGMI}^{2}$	*/*
D. Preamplifier	$E_{npre}^{2}$	$G_{pre}^{2}$
E. Multiplier	$E_{nmul}^{2}$	2 · $G_{mul}^{2}$
F. Filter	$E_{nfilter}^{2}$	$G_{filter}^{2}$
G. Instrumentation amplifier	$E_{nINA}^{2}$	$G_{INA}^{2}$

**Table 2 sensors-20-00960-t002:** Simulation parameters of the preamplifier.

	Parameter	$R_{8}$ Ω	$R_{9}$ Ω	$e_{n}$ nV/√Hz	$i_{n}$ pA/√Hz
Configurations					
1		1000 Ω	9100 Ω	2.6	2.7
2		100 Ω	910 Ω	2.6	2.7
3		100 Ω	910 Ω	1.2	2.8

**Table 3 sensors-20-00960-t003:** Noise contribution of each part under different configurations.

Configurations	Resistance	Operational Amplifier		Total
	$\mathrm{Thermal}\;\mathrm{Noise}\;\mathrm{Power}\;E_{t}^{2}$	$\mathrm{Voltage}\;\mathrm{Noise}\;\mathrm{Power}\;E_{vvi}^{2}$	$\mathrm{Current}\;\mathrm{Noise}\;\mathrm{Power}\;E_{ivi}^{2}$	$\mathrm{Total}\;\mathrm{Output}\;\mathrm{Noise}\;\mathrm{Power}\;E_{nvi}^{2}$
1	15.37 × 10^−16^ V^2^/Hz	6.92 × 10^−16^ V^2^/Hz	6.00 × 10^−16^ V^2^/Hz	28.30 × 10^−16^ V^2^/Hz
2	1.69 × 10^−16^ V^2^/Hz	6.86 × 10^−16^ V^2^/Hz	6.10 × 10^−18^ V^2^/Hz	8.64 × 10^−16^ V^2^/Hz
3	1.69 × 10^−16^ V^2^/Hz	1.46 × 10^−16^ V^2^/Hz	6.55 × 10^−18^ V^2^/Hz	3.20 × 10^−16^ V^2^/Hz

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
