# Peer review of "Noise Modeling and Simulation of Giant Magnetic Impedance (GMI) Magnetic Sensor"

_sensors, 2020, doi:10.3390/s20040960_

Round 1
Reviewer 1 Report
Dear Authors, Dear Editor,
The manuscript describes the circuit of a GMI based sensor used for sensing magnetic fields. The individual noise components are described and determined analytically. For one configuration of the sensor circuit also experimental results are compared to the analytical calculation.
I believe the work is relevant in general, however, the manuscript shows multiple flaws that should be addressed before publication.
A major point I would like to raise is in regard of the experimental part of the work. The analytical description of noise measurements is complex and some errors only become apparent with systematic study of the parameters of the system. As the authors only present a single experimental result, I am not convinced that the model correctly describes the system.
Would it be possible to include more experimental results in this study? For example, 2 additional V_g values could show a much more convincing agreement.
In addition, I have difficulties to understand the units used in this manuscript. Especially, the units of noise powers and gain parameters seem to deviate. Also some of the equations are not completely clear to me:
equation 1: Please check units and explain the origin of the factors in this equation.
equation 2: It seems this gain has a unit of 1/Ohm, while other Gains have no units. It is possible to change this so that all Gain values are of the same unit.
equation 4,5,6: It seems the unit of the noise spectral densities are A^2/Hz, but later in table 3 uses a different unit. Please check.
equation 14: This Gain has the unit of V. Is it possible to use a Gain without a unit instead?
equation 15: Is this equation complete? From the text it seems that something is missing here. I would be also interested in a typical value of E_mul.
equation 31/32: Should this be capital E and B? It also seems that equation 32 uses B_input instead of B_total.
Further I have to address the following points:
line 29: Reference 3-6 do not support the claim made in this sentence. If I count correctly all 4 references have in total 24 citations. Please, use more appropriate references to show world wide attention by researchers.
line 39-41: Please improve the structure of the sentence.
line 45: Which two conductors cause the 1/f noise?
line 325: I believe it is necessary to add a full table of the used parameters to the paper or supplemental materials. In addition, the authors could think about providing parts the matlab code in the supplementary information.
line 333 to 342: This part is partly duplicated in the conclusions. Please remove the duplicated sections here or in the conclusions.
line 335: What type is this noise? Why can it not be considered in the matlab estimation?
line 351: The expression "1/f frequency range" is unclear to me. What frequency range is meant here?
Reviewer 2 Report
Good work. English language and grammar have to be improved, especially concerning the correct use of articles.
Sec. 2.1.1.: explain "model AD9959".
Reviewer 3 Report
In this manuscript, the authors developed an analytical model to describe the equivalent magnetic noise for a giant magnetoimpedance (GMI) sensor. The circuits including an excitation source, a preamplifier, a multiplier, a low-pass filter and an instrument amplifier were investigated for theirs noise contributions to the total noise spectral density. The author compared their approaching results with the measurement and announced that:
The noise sources from the preamplifier and the multiplier dominates the white noise regime for GMI sensor. The 1/f noise is dominated by the noise source in the low pass filter. The obtained equivalent magnetic noise @ 1Hz was 5.66nT/rt-Hz.The results sounds interesting, but not convincing. Before the following questions are addressed, I cannot accept this manuscript to be published.
Question 1.
What is the novelty of this paper compared to previous published papers that have already described the noise contributions of GMI sensors? What are the sweet points of this proposed approaching method? Accuracy?
Question 2.
The author concluded that the 1/f noise is dominated by the noise source in the low-pass filter. But the measurements, in FIG 6, cannot support their conclusions. The large gap between the approaches and the measurement was attributed to the limit point number of the spectrum analyzer. This cannot convince me. Previous studies showed that the 1/f noise contribution was origined from the GMI sensor head or the excitation sources, but not from the detection circuit. If the authors insist that the low-pass filter provides the most noise contribution, better experimental results need to be shown to support their conclusions. At least, the low-pass filter with different operational amplifier should be tested, in order to check the change of the 1/f noise spectral density.
Question 3.
The given noise level of 5.66 nT/rt-Hz @1Hz was too high. It is even worse than the low-cost commercial HALL sensors. If the authors have no pressure to publish the paper fast, it is better to improve the noise performance first.
Question 4.
There are several strange reference, for example ref 9 is a paper on magnetoelectric (ME) sensors. But the authors cited it to describe the noise in GMI sensors. The circuits and noise behaviors are totally different between ME and GMI sensors.
Reviewer 4 Report
The article entitled “Noise modeling and simulation of Giant Magnetic Impedance (GMI) magnetic sensor” aims to evaluate the sources of noise in GMI sensors. The paper is well written. The synthetic model is well described and the simulation is convincing. However the article reads more as a good technical note than a scientific paper. It is not clear enough what is novelty and what is from the literature. I think the article would be good for publication if the authors would state more clearly and emphasize what is novel in the paper. A better explanation on why (and if it is significant or not) the experiment differs from the simulation is required (HF noise but also LF variations in the >1Hz range)
I provide hereafter more detailed comments:
Keywords: I do not know if Matlab is an important keyword
L 33-34 “the external interference.. so it is ignored. Please explain why or at list cite a reference for it
L. 35 please give values for middle and high frequencies
L. 37 please give values for low frequencies
L. 41 “2-3 order of magnitudes lower” please give values
L. 42 “Conditioning circuit is the main factor” => for low frequencies (give the range of low frequencies)
L. 53 “in Matlab” how? Did you use an existing script (reference) or you computed yourself a noise model (explain)
L. 59 obtained => computed
L. 257 1 Hz to 10 kHz, how does it copes with your definition of low, middle and high frequencies?
Fig 3 : please describe the variables in the caption , or include fig 2 which is really good in fig 3 (I have to jump to the fig 2 to understand fully fig 3)
Fig 4: please describe the variables in the caption (full name)
Round 2
Reviewer 1 Report
Dear Authors,
I believe that after the changes the manuscript can be published.
Best Regards.
Reviewer 3 Report
no
Reviewer 4 Report
The authors answered clearly to all of the concerns I had about the first version and the proposed changes make the manuscript very suitable for publication